# Parental opioid prescriptions and the risk of opioid use in adolescents and young adults: The HUNT Study linked with prescription registry data

Anna Marcuzzi[1,2]*, Paulo Ferreira[3], Paul Jarle Mork[1], Manuela L. Ferreira[4], Karoline Moe[1], Sigmund Gismervik[1,2], Anne Lovise Nordstoga[5], Tom Ivar Lund Nilsen[1]

1 Department of Public Health and Nursing, Norwegian University of Science and Technology (NTNU), Trondheim, Norway, 2 Clinic of Rehabilitation, St. Olavs Hospital, Trondheim, Norway, 3 Faculty of Medicine and Health, School of Health Sciences, Charles Perkins Centre, Musculoskeletal Health, The University of Sydney, Sydney, Australia, 4 Musculoskeletal Program, The George Institute for Global Health, University of New South Wales, Sydney, Australia, 5 Department of Neuromedicine and Movement Science, Norwegian University of Science and Technology (NTNU), Trondheim, Norway

* anna.marcuzzi@ntnu.no

## Abstract

### Background

Although opioids are usually not recommended for young people they are often prescribed. It is not clear whether family-level factors are related to the risk of opioid use among adolescents and young adults. The aim of this study is to examine the association between parental opioid prescriptions and risk of opioid use in young people.

### Methods and findings

A prospective cohort study, including 21,470 adolescents and young adults (13–29 years) participating in the third (2006–2008) or fourth (2017–2019) survey of the population-based Young-HUNT or HUNT Study, Norway, paired with at least one participating parent. Opioid prescriptions were obtained by a linkage to the Norwegian Prescription Database. Parents' opioid prescriptions were defined as '0', '1', and '≥2' prescriptions over a period of 5 years. Analyses were also stratified according to parental chronic musculoskeletal (MSK) pain status (no, yes) assessed by the Standardised Nordic Questionnaire. Two outcomes were assessed: 1) any opioid prescription, and 2) persistent opioid prescriptions (i.e., at least three out of four quarters of the year). Analyses were adjusted for parental age, parental education level, parental body mass index, offspring age, and offspring participation survey. Follow-up started at the date of survey participation and ended at the date of prescription, emigration/death, or 7-year follow-up. If the

which permits unrestricted use, distribution, and reproduction in any medium, provided the original author and source are credited.

**Data availability statement:** To protect participants' privacy, HUNT Research Centre aims to limit storage of data outside HUNT databank and cannot deposit data in open repositories. HUNT databank has precise information on data exported to this project and are able to reproduce these upon request. There are no restrictions regarding data export given approval of applications to HUNT Research Centre. For more information, see: http://www.ntnu.edu/hunt/data or contact the HUNT Study administration at kontakt@hunt.ntnu.no. Data from the national registries used in this study cannot be made publicly available without a specific approval from each registry.

**Funding:** A.M. was financially supported by Stiftelsen DAM (https://www.dam.no/; grant number 2022/FO387176) to carry out this work. The funder had no role in study design, data collection and analysis, decision to publish, or preparation of the manuscript.

**Competing interests:** The authors have declared that no competing interests exist.

**Abbreviations:** ATC, Anatomical Therapeutic Chemical; HR, hazard ratio; MSK, musculoskeletal; NorPD, Norwegian Prescription Database; STROBE, STrengthening the Reporting of OBservational studies in Epidemiology; CI, Confidence Interval.

mother or father had ≥2 opioid prescriptions, the hazard ratios (HR) for persistent opioid prescriptions in offspring were 2.60 (95% CI [1.86, 3.65]) and 2.37 (95% CI [1.56, 3.60]), respectively, compared to offspring whose parents did not receive any opioid prescriptions. There was no clear evidence that parental chronic MSK pain status influenced these associations. Comparing offspring of mothers with ≥2 versus no opioid prescriptions, the HR for any opioid prescription was 1.30 (95% CI [1.15, 1.47]) if the mother reported chronic MSK pain and 1.31 (95% CI [1.06, 1.62]) if she did not. Corresponding HRs associated with fathers' opioid prescription were 1.19 (95% CI [1.01, 1.41]) if the father reported chronic MSK pain and 1.21 (95% CI [0.98, 1.50]) if he did not. Residual confounding due to unmeasured factors cannot be excluded.

## Conclusions

Parental opioid prescription is related to an increased risk for opioid initiation and persistent use in offspring, irrespective of parental history of chronic MSK pain. These findings suggest that family-based strategies should be considered when managing pain and opioid use in young people.

## Author summary

### Why was this study done?

- Opioids are often prescribed for non-malignant pain despite potential adverse long-term consequences, particularly in young people.

- This study examined whether parental use of prescription opioids is associated with increased risk of opioid use in their offspring.

### What did the researchers do and find?

- This cohort study included 21,470 adolescents and young adults aged 13–29 years participating in the population-based Young-HUNT and HUNT Study in Norway paired with at least one parent. The Norwegian Prescription Registry was used to identify opioid prescriptions in parents and offspring.

- Cox proportional hazard regression was used to estimate hazard ratios for opioid prescription in offspring according to mothers' and fathers' opioid prescriptions.

- Offspring whose mother or father had two or more opioid prescriptions had higher risk of persistent opioid use in the subsequent 7 years compared to those whose parents did not receive any opioid prescription.

PLOS Medicine

### What do these findings mean?

- The study findings suggest that family-based strategies should be considered when managing pain conditions in adolescents and young adults to avoid potentially unnecessary opioid use.

- Residual confounding due to unmeasured factors influencing both parental and offspring opioid prescriptions cannot be ruled out.

## Introduction

Chronic pain is common among adolescents [1,2] and often continues as they transition into adulthood [3–6]. Pain in adolescents is often managed pharmacologically [7], despite the lack of evidence about its efficacy for non-malignant pain in this population [8,9]. In particular, prescription of strong analgesics such as opioids is not recommended for young people [10]. Norwegian data indicate a 10-year incidence of opioid prescription of 5% in adolescents (i.e., 13–19 years of age), rising to 36% in young adults (i.e., 20–32 years of age) [11]. While a large proportion of these prescriptions are single occurrences, e.g., for treating acute pain [12], they might nevertheless increase the risk of future persistent use and misuse [13].

Several factors have been associated with frequent opioid use in young people, including low socioeconomic status, diagnosis of multisite pain, and psychological challenges, e.g., mood disorders, illicit drug use, and childhood trauma [11,14]. However, family-level factors, such as parental opioid prescription and their relationship with the risk of opioid use in their offspring have not been extensively investigated. A previous retrospective cohort study showed an association between opioid use in a family member and persistent opioid use after surgery in adolescents and young adults living in the same household [15]. Similarly, based on self-reported data, a cross-sectional study showed a correlation between parental use of prescription opioids and adolescents' opioid use and misuse [16]. While these studies are based on US data and their generalisability to other contexts is unknown, they highlight that familial environmental risk factors may play an important role in adolescents' opioid use [16,17]. Furthermore, parental history of chronic musculoskeletal (MSK) pain has been previously linked with an increased risk of offspring's chronic pain [18–20], suggesting a genetic susceptibility to pain [21] that, in turn, may increase the risk of opioid use. It is therefore possible that parental chronic MSK pain status may influence the association between opioid prescriptions in parents and risk of opioid use in their adolescent and young adult offspring.

Since prescription of opioids has potentially adverse long-term consequences [13,22], a better understanding of familial factors in relation to opioid use in young people is important. This knowledge can guide healthcare providers in assessing individual risk when considering prescription of opioids for this age group. Therefore, the aims of this population-based study were to describe the overall incidence rates of opioid prescriptions in adolescents and young adults and prospectively examine whether opioid prescriptions in parents influence the risk of a first and also more persistent opioid prescriptions in this group of young people. We hypothesised that parental opioid prescriptions would be associated with an increased risk of offspring's opioid prescriptions as an indication of familial predisposition to opioid use.

## Methods

### Study sample

This cohort study consists of a sample of adolescents and young adults aged 13–29 years who participated in the third (2006–2008) or fourth (2017–2019) survey of the population-based Young-HUNT and HUNT Study in Norway. Each participants record in the HUNT Study was linked via unique personal identification number with 1) the family registry at Statistics Norway to identify parents also participating in the HUNT Study and 2) the Norwegian Prescription Database (NorPD) to obtain information on prescribed medications. All inhabitants aged 13–19 years residing in the North-Trøndelag

region in Norway were invited to the Young-HUNT Study, with 8,200 (78%) participating in Young-HUNT3 and 8,066 (76%) in Young-HUNT4 [23]. Correspondingly, all inhabitants aged 20 years or older were invited to the HUNT Study, with 50,807 (54%) participating in HUNT3 and 56,078 (54%) in HUNT4 [24]. A description of the HUNT participation and an overview of the various questionnaires can be found at https://www.ntnu.edu/hunt. This study was registered in the Open Science Framework Registry as part of a larger project [25] and is reported as per the STrengthening the Reporting of OBservational studies in Epidemiology (STROBE) guideline (S1 Checklist). Ethical approval was obtained by the Regional Committee for Medical and Health Research Ethics in Central Norway (Ref. 396895). All participants in the HUNT study signed an informed consent form for using their data for research.

### Linkage to the prescription database

The HUNT records were linked with the NorPD registry to identify parents' and offspring's medication prescription between January 2004 and August 2023. The NorPD contains information on all drug prescriptions dispensed to individuals from all pharmacies in Norway, whether they are reimbursed or not. Over the counter drugs, i.e., those purchased without a prescription or supplied to hospitals, nursing homes, or physicians' offices, are not included in the registry. All drugs in the NorPD registry are classified according to the Anatomical Therapeutic Chemical (ATC) classification system and reported along with defined daily dose values. Opioid prescriptions correspond to ATC group codes N02A. Additionally, the NorPD registry includes information on reimbursement codes and the prescriber's identification number [26]. Reimbursement codes are only issued for chronic conditions that require long-term medication (i.e., blue prescriptions) and thus, not all prescriptions have a reimbursement code. In this study, prescriptions with reimbursement codes indicating cancer diagnosis or palliative care (−50 and −90) were classified as prescriptions due to malignancy and were not defined as an outcome case among the offspring.

### Parental opioid prescriptions and chronic musculoskeletal pain

Any opioid prescription in parents occurring within 5 years of offspring's HUNT participation (i.e., 2 years before and 3 years after) was used as exposure variable. Offspring of parents who died or emigrated in this 5-year window were excluded from the analyses. Parental prescriptions were analysed separately for mothers and fathers and categorised as '0', '1', and '≥2'.

Information about parental history of chronic MSK pain was obtained from the Standardised Nordic Questionnaire [27] from the same HUNT survey as the offspring participation where available. If not available, an earlier survey was chosen, and, if not available, a later survey was chosen (shown in S1 Table). Mothers and fathers reporting 'yes' to the following question "During the last year, have you had pain and/or stiffness in muscles or joints that lasted at least 3 consecutive months?" were classified as having chronic MSK pain.

### Opioid prescriptions in offspring

Opioid prescriptions in offspring were classified as 1) any opioid prescription for non-malignant pain, and 2) persistent opioid prescriptions, defined as prescriptions for non-malignant pain occurring at least three out of four quarters of the year. The latter definition approximates the wide definition for persistent opioid use reported previously [28]. Follow-up started at the date of participation in the respective (Young-) HUNT survey and ended at the date of dispensing of opioids, emigration/death, or at 7-year follow-up (or at the end of the registry data in August 2023, if it occurred earlier). Prescriptions were included from the date of survey participation while ongoing prescriptions were not included unless they were renewed at or after the date of participation. In the subsample where both offspring and parents had an opioid prescription, the median [25th percentile, 75th percentile] days from a parental opioid prescription to an offspring prescription was 868 [−153, 2,467] for mothers and 1,004 [−92, 2,557] for fathers.

## Other variables

Parental age (year) at the time of offspring HUNT participation, parental highest level of education (≤12, >12 years), parental body mass index (kg/m$^2$), offspring age (year), and offspring participation survey (Young-HUNT3/HUNT3 or Young-HUNT4/HUNT4) were analysed as potential confounders (see directed acyclic graph in S1 Fig). We controlled for parental body mass index rather than offspring body mass index as this factor is more likely related to both the exposure (i.e., parental opioid prescription) and the outcome (offspring opioid prescription), the latter through, for example, offspring body mass index. All these variables were extracted from the HUNT data, except for parental education, which was retrieved from the National Education Database at Statistics Norway. Parental body mass index was retrieved following the same hierarchical structure as the parental MSK pain variable, i.e., the same HUNT survey as the offspring participation was used where available, and if not, an earlier survey was used, and finally a later survey was used if the earlier was not available (shown in S1 Table). Body mass index in mothers ($n = 423$) and fathers ($n = 373$), and parental education ($n = 23$) had missing data that were imputed (20 imputations) based on chained equations (*linear* for body mass index and *logit* for higher education) using all variables in the main analysis, including the outcome variable, as predictors in the imputation model.

## Statistical analysis

We estimated overall incidence rates per 1,000 person/years of any opioid prescriptions and persistent opioid prescriptions in offspring and present these according to ages 13–19, 20–24, 25–29, and 30–36 years during follow-up. Linear trends in opioid prescription rates were estimated from least-squares linear regression across the age groups and weighted by the inverse variance of the incidence rate [29]. Cox proportional hazard regression was used to estimate hazard ratios (HRs) for opioid prescription (i.e., any prescription and persistent prescriptions) with 95% confidence intervals (CI) in offspring according to mothers' and fathers' opioid prescriptions. Proportional hazard assumptions were assessed both graphically (log–log plots) and by test of Schoenfeld residuals, and no violations were observed. To address potential confounding by indication caused by increased susceptibility to pain in offspring of parents with chronic MSK pain, the association between parental opioid prescription (no prescription versus ≥2 prescriptions) and risk of any opioid prescription in offspring was analysed both as a joint variable with parental chronic MSK pain (no/yes) as well as stratified according to parental MSK pain. These analyses were conducted separately for mothers and fathers, and 3,075 and 5,724 offsprings were excluded due to missing information on mothers' or fathers' chronic MSK pain, respectively. A higher proportion of mothers and fathers had opioid prescriptions if they had missing data on chronic pain compared to those without missing data (S2 Table). Due to the low numbers of cases with persistent opioid prescription, the joint and stratified analyses were conducted only on the risk of any opioid prescription.

To assess the robustness of these results, we conducted several supplementary sensitivity analyses. To assess the possible influence of the timing of exposure assessment, we conducted two sensitivity analyses: 1) including only parental opioid prescriptions in the 2-year period prior to the start of follow-up and 2) including only parental opioid prescriptions in the 3-year period after the start of follow-up. In this analysis, parental opioid prescriptions were categorised as yes/no (Table 4). Then, we restricted the sample to offspring where information on both parents was available (i.e., trios) to evaluate possible selection bias (S3 and S4 Tables). Furthermore, the main analysis was also stratified according to offspring age (±20 years) to assess if the associations differed according to how likely the offspring were to live with their parents (S5 Table). The joint effect of parental opioid prescriptions and chronic MSK pain on offspring risk of opioid prescriptions was also restricted to parent-offspring associations assessed at the same HUNT survey (S6 Table). Also, we excluded offspring with any opioid prescription1 year prior to the survey participation (i.e., start of follow-up) to avoid influence of prevalent opioid use (S7 Table). All main and supplementary analyses were pre-planned, with the exception of those presented in Tables 4 and S7 which were included after peer-review.

Statistical analyses were performed using Stata 18.0 (StataCorp LLC, Stata Statistical Software: release 18. College Station, TX, USA).

## Results

Fig 1 shows the selection of participants into the study. Overall, 21,470 adolescents and young adults were included (mean age [SD]: 18.0 [4.4], females: 10,581 [52.8%]), of which 20,081 (93.5%) had complete information on mothers and 17,773 (82.8%) had complete information on fathers. The descriptive characteristics of the sample stratified by mother's and father's opioid prescription status (yes/no) are reported in Table 1.

A total of 5,232 (24.4%) adolescents and young adults had at least one opioid prescription during follow-up, and 270 (1.3%) had persistent opioid prescription. For the first opioid prescription (Fig 2A), the incidence rate (per 1,000 person/years) increased from 31.8 (95% CI [30.4, 33.4]) in adolescents (≤19 years) to 51.3 (95% CI [49.1, 53.5]) in those between 20–24 years and then further increased to 64.2 (95% CI [58.9, 70.0]) at age 30–37 years. This corresponded to a 4.9% (95% CI [−0.6, 10.4]; $p = 0.06$) increase for each 5-year age group. For persistent opioid prescriptions (Fig 2B), the incidence rate (per 1,000 person/years) increased from 1.0 (95% CI [0.8, 1.4]) in adolescents (≤19 years) to 2.6 (95% CI [2.1, 3.2]) in young adults (20–24 years old) and up to 5.6 (95% CI [4.2, 7.4]) in those 30–37 years, corresponding to a 12.1% (95% [3.8, 20.4]; $p = 0.02$) increase for each 5-year age group.

If mothers received ≥2 opioid prescriptions, the adjusted HRs in offspring were 1.34 (95% CI [1.21, 1.48]) for any opioid prescription and 2.60 (95% CI [1.86, 3.65]) for persistent opioid prescriptions, compared to offspring whose mothers did not receive any opioid prescriptions. The corresponding father-offspring associations gave HRs of 1.19 (95% CI [1.05, 1.34]) and 2.37 (95% CI [1.56, 3.60]), respectively (Table 2).

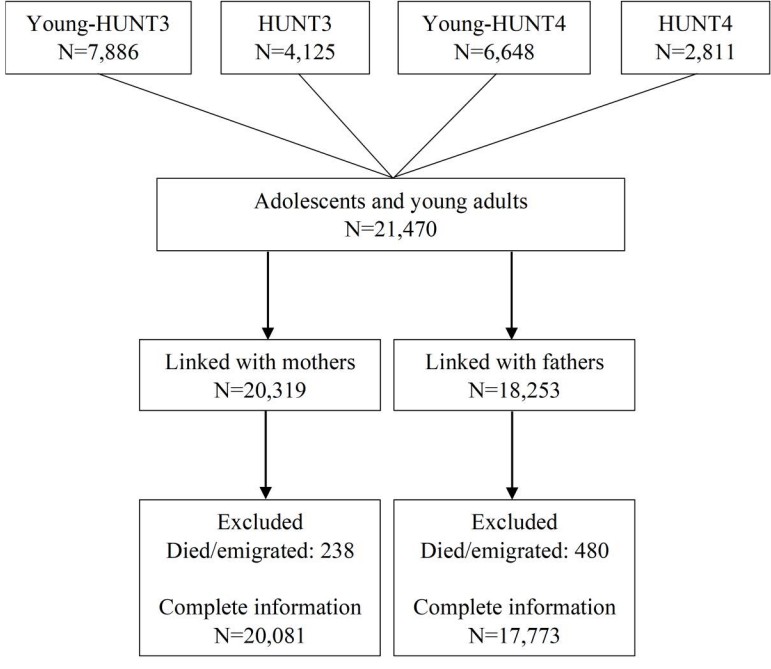

**Fig 1. Participants' flow chart.**

**Table 1. Descriptive characteristics of the sample stratified by number of opioid prescriptions in their mothers and fathers.**

| Characteristic | Mothers' opioid prescriptions | | Fathers' opioid prescriptions | |
|---|---|---|---|---|
| | No | Yes | No | Yes |
| No. of adolescents/young adults | 16,923 | 3,158 | 15,351 | 2,422 |
| Age, mean (SD), years | 18.0 (4.3) | 17.9 (4.4) | 18.0 (4.3) | 18.0 (4.4) |
| Females, n (%) | 8,922 (52.7) | 1,678 (53.2) | 7,999 (52.1) | 1,302 (53.8) |
| Parental age, mean (SD), years | 46.2 (6.2) | 45.9 (6.6) | 49.2 (6.6) | 49.6 (6.8) |
| Parental education, n (%) | | | | |
| <10 years | 2,020 (12.0) | 426 (13.6) | 1,892 (12.3) | 329 (13.6) |
| 10–12 years | 7,317 (43.3) | 1,462 (46.5) | 9,244 (60.3) | 1,504 (62.2) |
| >12 years[a] | 7,551 (44.7) | 1,257 (39.9) | 4,197 (27.4) | 586 (24.2) |
| Parental BMI, mean (SD), kg/m² | 26.5 (4.8) | 27.3 (5.1) | 27.2 (3.8) | 28.1 (4.0) |

[a]Higher education.

Abbreviations: SD, standard deviation; BMI, body mass index.

The joint and stratified effects of parental use of prescription opioids and parental chronic MSK pain on any opioid prescription in offspring are reported separately for mothers and fathers (Table 3). Although offspring had an HR of 1.51 (95% CI [1.34, 1.70]) of any opioid prescription if mothers had both ≥2 opioid prescriptions and reported chronic MSK pain, compared to offspring of mothers who did not have any opioid prescription and reported no chronic MSK pain, there was no evidence of effect modification. The offspring's HRs of any opioid prescriptions if mothers had ≥2 opioid prescriptions versus no prescriptions were similar whether mothers reported chronic MSK pain or not (1.30, 95% CI [1.15, 1.47] and 1.31, 95% CI [1.06, 1.62], respectively). The corresponding analysis of father-offspring associations gave HRs of 1.23 (95% CI [1.05, 1.45]), 1.19 (95% CI [1.01, 1.41]), and 1.21 (95% CI [0.98, 1.50]), respectively.

Using only parental opioid prescriptions from either before or after the start of follow-up gave largely similar HRs as in the main analysis (Table 4). Restricting the study sample to parent-offspring associations where both parents provided data resulted in largely similar HRs as described above, both for any opioid prescriptions and for persistent opioid prescriptions (S3 and S4 Tables). Furthermore, the HRs of opioid prescriptions in offspring were largely similar when stratifying the analysis by offspring age, i.e., <20 years and ≥20 years (S5 Table). In S6 Table, the joint effect of parental opioid prescriptions and chronic MSK pain on offspring risk of opioid prescriptions was restricted to parent-offspring associations assessed at the same HUNT survey, showing similar HRs as for the full sample. In S7 Table, the HRs of opioid prescriptions in offspring were similar to those of the main analysis, after excluding offspring who had an opioid prescription 1 year prior to survey participation.

## Discussion

In this study, nearly a quarter of adolescents and young adults (13–29 years old) had at least one opioid prescription for non-malignant pain during the 7-year follow-up, while a small proportion had persistent opioid use during that period. The incidence rate of persistent opioid prescriptions increased linearly with age. We found that opioid prescription in parents was associated with an increased risk of persistent opioid use in their offspring. Furthermore, young adults whose parents had both a history of chronic MSK pain and had ≥2 opioid prescriptions experienced the highest risk of any opioid prescription, although there was no evidence of a synergistic effect of these parental factors.

Our finding of increased risk of persistent opioid use in offspring of parents who used prescription opioids strengthens the evidence from previous studies reporting a positive association of parental and also other family members' opioid use with adolescents' opioid use, although it is not possible to directly compare estimates due to different designs and

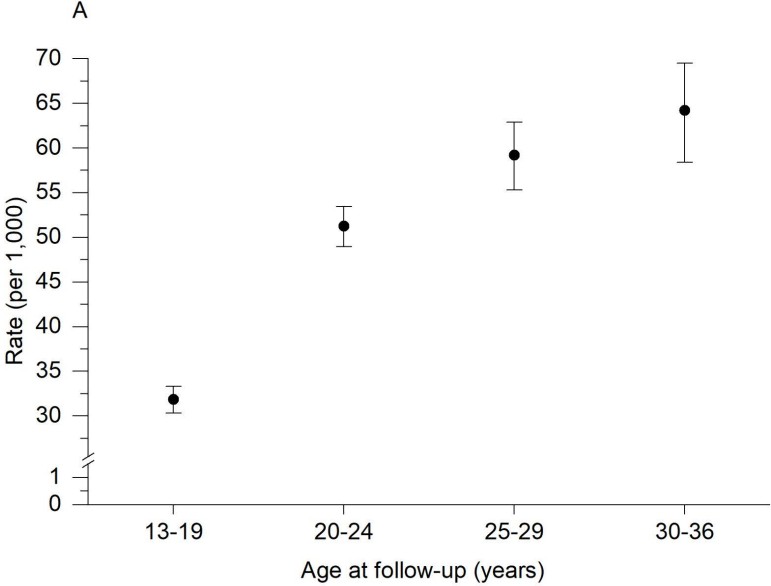

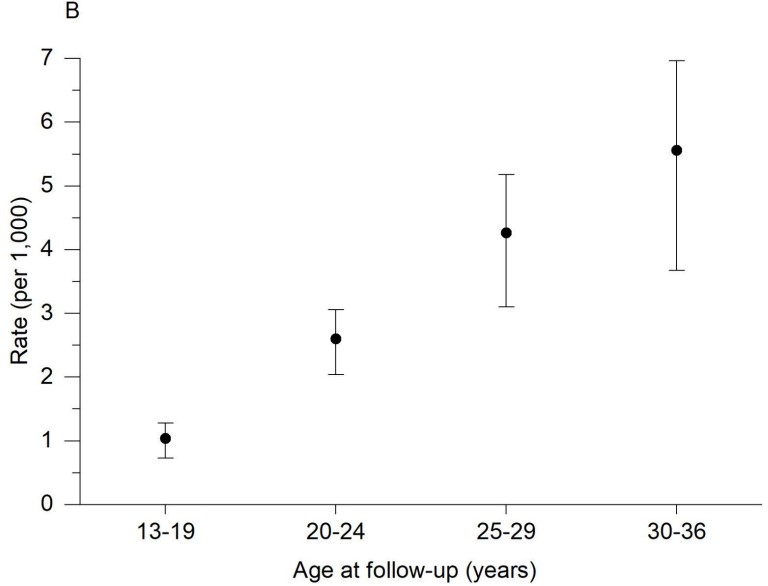

**Fig 2. Incidence rate per 1,000 person/years with 95% confidence intervals of any opioid prescription (A) and persistent opioid prescription (B) by age groups at follow-up.**

methods [15,16]. Several mechanisms may mediate this association, including parental norms and preferences regarding the use of analgesics [30], shared health-seeking behaviour influenced by familial vulnerability to illnesses, role modelling, and broader family context [31]. Similarly, clinician preferences in prescribing practices may also contribute to this association [16,17], particularly in the Norwegian context, where it is common that parents and adolescents share the same general practitioner [32]. Variations in clinical practice may also influence preferences towards a particular general

**Table 2. Number of parental opioid prescriptions and risk of opioid prescriptions in offspring.**

| No. of parental prescriptions | Any opioid prescription | | | | Persistent opioid prescriptions[a] | | | |
|---|---|---|---|---|---|---|---|---|
| | Person years | No. of cases | Crude HR [95% CI] | Adjusted[b], HR [95% CI] | Person years | No. of cases | Crude HR [95% CI] | Adjusted[b], HR [95% CI] |
| Mothers | | | | | | | | |
| 0 | 94,066 | 3,975 | 1.00 [reference] | 1.00 [reference] | 95,413 | 183 | 1.00 [reference] | 1.00 [reference] |
| 1 | 9,214 | 428 | 1.12 [1.01,1.24] | 1.12 [1.01,1.23] | 9,360 | 16 | 0.92 [0.55,1.54] | 0.91 [0.55,1.52] |
| ≥2 | 7,598 | 442 | 1.40 [1.27,1.55] | 1.34 [1.21,1.48] | 7,767 | 42 | 2.91 [2.08,4.07] | 2.60 [1.86,3.65] |
| Fathers | | | | | | | | |
| 0 | 85,602 | 3,662 | 1.00 [reference] | 1.00 [reference] | 86,831 | 168 | 1.00 [reference] | 1.00 [reference] |
| 1 | 7,748 | 340 | 1.05 [0.94,1.17] | 1.04 [0.93,1.16] | 7,844 | 12 | 0.81 [0.45,1.46] | 0.80 [0.45,1.44] |
| ≥2 | 5,299 | 275 | 1.24 [1.09,1.40] | 1.19 [1.05,1.34] | 5,435 | 26 | 2.51 [1.66,3.80] | 2.37 [1.56,3.60] |

[a]Defined as prescription of opioids in at least three quarters of the year.

[b]Adjusted for parental age at time offspring participated in HUNT survey (continuous), parental highest education (<12, ≥12 years), parental body mass index (continuous), offspring age (continuous) and survey of offspring's participation (Young-HUNT3/HUNT3, Young-HUNT4/HUNT4).

Abbreviations: HR, hazard ratio; CI, confidence interval.

**Table 3. Joint effect of parental opioid prescription and chronic musculoskeletal pain on risk of any opioid prescription in offspring.**

| Parental chronic MSK pain | No opioid prescriptions | | | | ≥2 opioid prescriptions | | | | Within musculoskeletal pain strata estimates |
|---|---|---|---|---|---|---|---|---|---|
| | Person years | No. of cases | Crude, HR [95% CI] | Adjusted[a], HR [95% CI] | Person years | No. of cases | Crude, HR [95% CI] | Adjusted[a], HR [95% CI] | Adjusted[a], HR [95% CI] |
| Mothers | | | | | | | | | |
| No | 44,085 | 1,683 | 1.00 [reference] | 1.00 [reference] | 1,835 | 92 | 1.36 [1.11, 1.68] | 1.29 [1.05, 1.59] | 1.31 [1.06, 1.62] |
| Yes | 41,499 | 1,911 | 1.16 [1.09, 1.24] | 1.15 [1.08, 1.23] | 5,152 | 314 | 1.58 [1.40, 1.78] | 1.51 [1.34, 1.70] | 1.30 [1.15, 1.47] |
| Fathers | | | | | | | | | |
| No | 43,411 | 1,784 | 1.00 [reference] | 1.00 [reference] | 1,801 | 90 | 1.28 [1.03, 1.58] | 1.22 [0.99, 1.51] | 1.21 [0.98, 1.50] |
| Yes | 30,734 | 1,360 | 1.04 [0.97, 1.12] | 1.04 [0.97, 1.11] | 2,972 | 159 | 1.28 [1.09, 1.51] | 1.23 [1.05, 1.45] | 1.19 [1.01, 1.41] |

[a]Adjusted for parental age at time offspring participated in HUNT survey (continuous), parental highest education (<12, ≥12 years), parental body mass index (continuous), offspring age (continuous) and survey of offspring's participation (Young-HUNT3/HUNT3, Young-HUNT4/HUNT4).

Abbreviations: MSK, musculoskeletal; HR, hazard ratio; CI, confidence interval.

**Table 4. Sensitivity analysis on parental opioid prescription (yes/no) and risk of any opioid prescription in offspring, including parental prescriptions either before or after start of follow-up.**

| Parental prescription | Parental prescriptions 2 years prior start of follow-up | | | | Parental prescriptions 3 years after start of follow-up | | | |
|---|---|---|---|---|---|---|---|---|
| | Person years | No. of cases | Crude, HR [95% CI] | Adjusted[a], HR [95% CI] | Person years | No. of cases | Crude, HR [95% CI] | Adjusted[a], HR [95% CI] |
| Mother | | | | | | | | |
| No | 107,082 | 4,632 | 1.00 [reference] | 1.00 [reference] | 96,211 | 4,070 | 1.00 [reference] | 1.00 [reference] |
| Yes | 3,790 | 213 | 1.38 [1.21, 1.59] | 1.24 [1.08, 1.43] | 14,668 | 775 | 1.27 [1.18, 1.37] | 1.25 [1.16, 1.35] |
| Father | | | | | | | | |
| No | 95,643 | 4,129 | 1.00 [reference] | 1.00 [reference] | 87,574 | 3,751 | 1.00 [reference] | 1.00 [reference] |
| Yes | 3,006 | 148 | 1.20 [1.02, 1.41] | 1.09 [0.93, 1.29] | 11,075 | 526 | 1.12 [1.02–1.23] | 1.11 [1.01–1.22] |

[a]Adjusted for parental age at time offspring participated in HUNT survey (continuous), parental highest education (<12, ≥12 years), parental body mass index (continuous), offspring age (continuous), and survey of offspring participation (Young-HUNT3/HUNT3, Young-HUNT4/HUNT4).

Abbreviations: HR, hazard ratio; CI, confidence interval.

practitioner which could further reinforce this association [31]. Several studies have also indicated that other environmental factors such as opioid availability in the household might be important for opioid initiation, long-term opioid use and misuse among family members [17,33,34]. Our analysis stratified by offspring age (i.e., <20 years, ≥20 years) did not show any substantial difference in the estimates between age groups, suggesting that living in the same household with parents did not have a differential effect on the risk of opioid prescription in offspring. However, offspring age may not fully capture family living conditions, both in terms of when offspring move out of the family home and due to more complex family structures, particularly in families experiencing greater burden (e.g., chronic pain and opioid use) [35].

Our study also shows that young people whose parents had both a history of chronic MSK pain and ≥2 opioid prescriptions were those at highest risk for a first opioid prescription compared to those with only one parental exposure. Notably, parental history of chronic MSK pain alone, particularly in mothers, was associated with an increased risk of offspring opioid use. As chronic MSK pain has been shown to cluster in families [36], it could be speculated that genetic influences interacting with other psychosocial factors, e.g., inherited pain-related traits, catastrophising, pain-specific social learning, general parenting, and family health [21,37,38], could lead to increased pain vulnerability potentially translating into higher treatment burden. Nonetheless, opioid prescriptions in parents showed largely the same association with the risk of offspring's opioid prescription, irrespective of parental history of MSK pain, suggesting that potential familial vulnerability to pain does not modify the parent-offspring association in opioid use.

Although persistent use of opioids was relatively rare in this study sample, receiving at least one opioid prescription was more common, particularly in late adolescence and young adulthood, as shown by the increasing incidence rates with age. It is possible that this increase can be related to more restrictive prescription practices in younger adolescents. Clinical practice recommendations advise cautious use of prescription opioids (e.g., initially optimising non-pharmacological and/or non-opioid analgesic treatments) [39], particularly in young people [10,40]. Thus, the strong association between parent and offspring opioid prescription observed in this study highlights the need for increased awareness among health personnel that parental opioid use is important to consider when managing pain conditions in adolescents and young adults. For example, history of familial opioid prescription could help identify patients with particularly high risk of prolonged opioid use in a clinical setting. From a public health perspective, the value of family-based preventive strategies, and, more broadly, multilevel interventions (e.g., targeting both the family and the clinicians) should be explored to increase efforts in preventing potentially unnecessary opioid use in young people [41]. This would also require a better understanding of processes underlying social interactions, e.g., among family members, friends, healthcare personnel, and their influence on individual patients' decision-making [42].

The strengths of this study are the prospective design, the large sample of young people, and that information on opioid prescriptions in parents and offspring was based on nationwide registry data in Norway. Some limitations need to be acknowledged. First, given that offspring were included if they had at least one parent participating in the HUNT survey, it is possible that the study comprised a healthier sample compared to non-responders [43] that could contribute to underestimate the parent-offspring associations of opioid use. Although the results from this study may be particularly relevant in a Norwegian setting, the findings may transport to other populations with similar family and healthcare structures. Furthermore, we did not categorise parental persistent opioid use with the same definition as offspring opioid use due to few cases and lack of statistical power. Additionally, the 5-year window to identify parental opioid prescription was chosen arbitrarily, but with an ambition to capture opioid use in parents around the time of offspring's participation. Events occurring outside this time window could potentially have influenced the offspring's risk of opioid use. In addition, parental opioid prescriptions were measured both before and after offspring HUNT participation. This may increase the likelihood that some parental opioid prescriptions in the 3-year period after start of follow-up could be related to previous offspring opioid use (i.e., reverse causation). However, sensitivity analysis separating parental opioid prescriptions occurring either before or after start of follow-up yielded largely similar estimates as the main results. Furthermore, familial liability to opioid use may exert its effect irrespective of the timing of opioid prescriptions.

It is also possible that some offspring had opioid prescriptions before their entry into the study, and this might have resulted in underestimated associations. However, we conducted a sensitivity analysis excluding offspring with opioid prescriptions 1 year prior to study participation, and the results remained similar to the main analyses. Parental chronic MSK pain was not necessarily reported at the time of offspring participation, although our sensitivity analysis, including only offspring where parental chronic MSK pain was assessed in the same survey as the offspring participation showed comparable results. Missing data on chronic MSK pain in parents was not imputed since this variable was used as a stratum classification variable. While parental age was similar between those with and without data on parental MSK pain, the proportion of parents with opioid prescription was larger in the latter group (S2 Table). This may contribute to selection bias, thus underestimating the joint association of parental pain and parental opioid prescriptions with offspring opioid prescriptions. We cannot rule out residual confounding due to unmeasured factors, e.g., family socioeconomic status, parental chronic diseases and other substance use influencing both their own and their offspring's risk of opioid prescriptions. Controlling for differences in socioeconomic status might be of particular importance, but indicators such as income or area-based socioeconomic level were either not available or not relevant in this homogenous and relatively rural setting. However, the complete registry-based education history available in this study is likely to capture important variation in socioeconomic status in the Norwegian context.

In conclusion, the results of this study show that opioid prescription in parents is related to an increased risk of opioid initiation and persistent opioid use in adolescents and young adults, irrespective of parental history of chronic MSK pain. These findings suggest that family-based strategies, targeting both parents and their offspring, should be considered when managing pain and reducing potentially unnecessary opioid use in young people.

## Supporting information

**S1 Table. Survey used for parental chronic MSK pain and body mass index variables.**
(DOCX)

**S2 Table. Parental age and opioid prescription stratified by missingness in parental chronic musculoskeletal pain variable.**
(DOCX)

**S3 Table. Effect of parental opioid prescription on risk of any opioid prescription in offspring for the full sample and restricted sample with information on both parents (i.e., trios).**
(DOCX)

**S4 Table. Effect of parental opioid prescription on risk of persistent opioid prescription in offspring for the full sample and restricted sample with information on both parents (i.e., trios).**
(DOCX)

**S5 Table. Effect of parental number of opioid prescriptions on risk of any opioid prescription in offspring stratified by offspring age.**
(DOCX)

**S6 Table. Joint effect analysis restricted to offspring whose parents have information about chronic musculoskeletal pain at the same survey.**
(DOCX)

**S7 Table. Number of parental opioid prescriptions and risk of opioid prescriptions in offspring removing those who had opioid prescription 1-year prior to participation.**
(DOCX)

**S8 Table. Descriptive characteristics of the sample stratified by number of opioid prescriptions in their mothers and fathers.**
(DOCX)

**S1 Fig. Directed acyclic graph.** Variables in black boxes are confounders that are adjusted for in the analyses. To avoid potential residual confounding by offspring age, this variable was also used as a confounder in the analyses. Abbreviations: MSK, musculoskeletal; BMI, body mass index; SES, socioeconomic status.
(DOCX)

**S1 Checklist. STrengthening the Reporting of OBservational studies in Epidemiology (STROBE) Statement—checklist of items that should be included in reports of observational studies, available at https://www.strobe-statement.org/, licenced under CC BY 4.0 Copyright 2025.**
(DOCX)

## Acknowledgments

The Trøndelag Health Study (HUNT) is a collaboration between HUNT Research Centre (Faculty of Medicine and Health Sciences, Norwegian University of Science and Technology NTNU), Trøndelag County Council, Central Norway Regional Health Authority, and the Norwegian Institute of Public Health.

## Author contributions

**Conceptualization:** Anna Marcuzzi, Paulo Ferreira, Paul Jarle Mork, Karoline Moe, Tom Ivar Lund Nilsen.

**Data curation:** Anna Marcuzzi, Karoline Moe, Anne Lovise Nordstoga.

**Formal analysis:** Anna Marcuzzi, Tom Ivar Lund Nilsen.

**Funding acquisition:** Anna Marcuzzi, Tom Ivar Lund Nilsen.

**Investigation:** Anna Marcuzzi, Paulo Ferreira, Paul Jarle Mork, Manuela L. Ferreira, Sigmund Gismervik, Tom Ivar Lund Nilsen.

**Methodology:** Anna Marcuzzi, Paulo Ferreira, Paul Jarle Mork, Manuela L. Ferreira, Karoline Moe, Sigmund Gismervik, Anne Lovise Nordstoga, Tom Ivar Lund Nilsen.

**Project administration:** Anna Marcuzzi, Tom Ivar Lund Nilsen.

**Resources:** Anne Lovise Nordstoga.

**Supervision:** Paulo Ferreira, Paul Jarle Mork, Tom Ivar Lund Nilsen.

**Visualization:** Paul Jarle Mork.

**Writing – original draft:** Anna Marcuzzi.

**Writing – review & editing:** Anna Marcuzzi, Paulo Ferreira, Paul Jarle Mork, Manuela L. Ferreira, Karoline Moe, Sigmund Gismervik, Anne Lovise Nordstoga, Tom Ivar Lund Nilsen.

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
