## [Editor Report · Decision Letter 0]

31 Jan 2025

Dear Dr Marcuzzi,

Thank you for submitting your manuscript entitled "Parental opioid prescriptions and the risk of opioid use in adolescents and young adults: The HUNT Study linked with prescription registry data" for consideration by PLOS Medicine.

Your manuscript has now been evaluated by the PLOS Medicine editorial staff as well as by an academic editor with relevant expertise and I am writing to let you know that we would like to send your submission out for external peer review.

Please re-submit your manuscript within two working days, i.e. by Feb 04 2025 11:59PM.

Kind regards,

Suzanne De Bruijn, PhD

Associate Editor

PLOS Medicine

---

## [Decision Letter · Decision Letter 1]

2 May 2025

Dear Dr Marcuzzi,

Many thanks for submitting your manuscript "Parental opioid prescriptions and the risk of opioid use in adolescents and young adults: The HUNT Study linked with prescription registry data" (PMEDICINE-D-25-00353R1) to PLOS Medicine. The paper has been reviewed by subject experts and a statistician; their comments are included below and can also be accessed here: [LINK]

As you will see, the reviewers think your study addresses an important question. However, they also had some concerns, specifically about potential selection bias, and the influence of certain assumptions. After discussing the paper with the editorial team and an academic editor with relevant expertise, I'm pleased to invite you to revise the paper in response to the reviewers' comments. We plan to send the revised paper to some or all of the original reviewers, and we cannot provide any guarantees at this stage regarding publication.

We ask that you submit your revision by May 23 2025 11:59PM. However, if this deadline is not feasible, please contact me by email, and we can discuss a suitable alternative.

Don't hesitate to contact me directly with any questions (sbruijn@plos.org).

Best regards,

Suzanne

Suzanne De Bruijn, PhD

Associate Editor

PLOS Medicine

sbruijn@plos.org

Comments from the reviewers:

Reviewer #1: Marcuzzi and team utilize a large cohorts and data linkage to explore the associations between parental opioid use and subsequent offspring opioid use. This is an important question that may offer insights into targeted interventions for reducing opioid use in the community.

Main queries for authors:

1. How were the linkages performed? Was this via unique identifying number and parent/child identifier or probabilistic matching? If the latter, what were the proportion of successful linkages?

2. Given a Cox model was used was the proportional hazard assumption checked? This should be included.

3. 95% confidence interval are missing for all crude values in the tables, these must be included.

4. The authors have assumed that those without the reimbursement code for opioid use were for non-malignant pain and were included. Is this assumption based on anything in terms of reimbursement reporting? How many had a missing reimbursement code? This is a significant assumption that may alter the included population.

5. Why were those with missing BMI data excluded? Was imputation considered for this variable?

6. It is likely that those with missing information on pain may not be missing at random and therefore excluding these cases may introduce bias. The authors should consider imputation or sensitivity analyses exploring this.

7. The persistent use and number of prescriptions should be consistent, it is confusing to switch between >3/4 of the year and >2 prescriptions.

8. Adjusted analyses: How were covariates selected? Were direct acyclic graphs used? If so, please include this.

9. Were other variables of socioeconomic position considered such as postcode to capture more deprived areas, household wealth, alcohol and substance use etc. Given the interrelatedness of SES and opioid use/misuse fully capturing this is critical, it is plausible that familial deprivation is driving this association.

10. Why was parental BMI selected and not offspring BMI? BMI has been associated with opioid use, thus this should be focused on the child. Is this not available and parental BMI a proxy for this? If so this should be stated.

11. For the increase in opioid use with offspring age, was this statistically different? If not, this should be included. Additionally, caution is needed when interpreting this result, this will likely in some part be due to provider preference to avoid opioids in minors, especially for persistent use.

12. Age </>20 does not seem the best surrogate for living in the home, especially in setting with complex families (opioid use and chronic pain) and to me is insufficient to suggest that 'contextual factors' might be less relevant (line 248).

13. BMI does not present as a negative control for me; BMI is associated with opioid use and offspring and parental BMI are linked. Unsure whether this analysis is additive to the manuscript.

Minor:

1. Table 1: suggest making simpler with no opioid vs any opioid use and putting the stratified into the supps

2. Suggest toning down of language -line 235, rather was associated with?

3. Line 74 - secondary aim could be removed from introduction

4. Line 61 simpler to say confounding by indication? And be clarified throughout that pain was included to investigate confounding by indication.

5. Psychological challenges - line 52 to be clarified

Reviewer #2: This paper addresses an important topic. Strengths include linkage with the Norway prescription drug database; however, the low participation in the Hunt study (~54%) raise concerns about generalizability and potential selection bias.

1. My main concern for this study is the potential for selection bias. For example, there may be immortal time bias as parents may fill a prescription on some date but the survey is completed sometime later - and yet follow up starts not at the time of the prescription fill but at survey completion, i.e., the offspring are "immortal" between the exposure (prescription) and survey completion. Additionally, participation in the Hunt study is very low. What is the population that is represented by the participants in this study? How generalizable are the findings?

2. I'm unclear why parental BMI is a confounder in this study. For a variable to be a confounder, it must be a common cause of the exposure and outcome. How is parental BMI a cause of opioid prescriptions in offspring?

3. The authors raise the issue of offspring opioid use outside the time window. Why not apply a washout period (e.g., 6 months - 1 year) to ensure that the offspring in your sample had no prior prescriptions?

4. It seems that there is missing data with respect to the reimbursement code for some prescriptions. What is the prevalence of missingness? The ad hoc method to classify "prescriptions with no reimbursement code … as prescriptions due to non-malignant pain" can introduce bias depending on the degree of missingness.

5. There are some illogical statements in the Discussion. For example, the authors state "Thus, the strong association between parent and offspring opioid prescription observed in this study highlights the need for increased awareness among health personnel that familial factors are important to consider when prescribing opioids as part of managing pain conditions in adolescents and young adults." But this study looked at prescribing opioids to parents, so this conclusion is not supported.

Reviewer #3: A useful contribution to an accumulating body of evidence about family (parental) prescription opioid use and its interesting connection with offspring prescription opioid use. The study usefully builds upon prior evidence in several ways. First, the predictors and outcome are conceptualized as receipt of prescriptions for an opioid compound -- not harmful or hazardous use, and possibly not even use, to the extent that some opioid prescriptions are issued/filled but then the compound is not taken. Second, there is a clearly specified epidemiologically credible population under study with a well-described sample of sufficient size to yield reasonably precise hazard ratio estimates. Third, the study design is prospective, whereas most evidence is based on non-prospective designs. Fourth, the analysis and estimation approaches are described with clarity.

The authors have an opportunity to say more about potential public health tactics and what clinicians might do in office-based practices. It is not clear that many readers will be surprised to find that there is some degree of within-family interdependence of medicine-seeking outcomes, although the focus on prescription-only opioids use is relatively novel (i.e., in contrast with the many studies of opioids used outside of the boundaries of clinically prescribed instructions).'

Some value might be gained by considering the early sociological studies of diffusion of prescribing practices -- i.e., interdependencies within practice settings -- and work in the direction of treatment choices. This body of evidence generally focuses upon the behavior of clinician-prescribers and the choices they make on behalf of their patients. The conceptual models that guided these studies of clinicians might prove to offer guidance for conceptual models needed to interpret this study's interesting estimates. At first blush, the mechanism of behavioral imitation might come to mind, but the choice models and discussions of these types of interdependencies offer more intriguing possibilities (e.g., see Manski on treatment choice and the difficulties faced when distinguishing what might be called 'social effects' from other sources of interdependencies).

Some word choices might be re-considered. For example, the abstract starts by seeming to promise causal inference, but the reader will be disappointed if the expectation is definitive evidence strong enough to support causal inference.

---

* Please upload any figures associated with your paper as individual TIF or EPS files with 300dpi resolution at resubmission; please read our figure guidelines for more information on our requirements: http://journals.plos.org/plosmedicine/s/figures. While revising your submission, please upload your figure files to the PACE digital diagnostic tool, https://pacev2.apexcovantage.com/. PACE helps ensure that figures meet PLOS requirements. To use PACE, you must first register as a user. Then, login and navigate to the UPLOAD tab, where you will find detailed instructions on how to use the tool. If you encounter any issues or have any questions when using PACE, please email us at PLOSMedicine@plos.org.

* Please ensure that the study is reported according to the STROBE guideline and include the completed STROBE checklist as Supporting Information. When completing the checklist, please use section and paragraph numbers, rather than page numbers. Please add the following statement, or similar, to the Methods: "This study is reported as per STROBE guideline (S1 Checklist)."

SUPPLEMENTARY MATERIAL

REFERENCES

[STUDY TYPE-SPECIFIC REQUESTS - DELETE SECTIONS AS NECESSARY]

OBSERVATIONAL STUDIES

* Abstract: Please include the study design, population and setting, number of participants, years during which the study took place (enrollment and follow up), length of follow up, and main outcome measures.

* Please ensure that the study is reported according to the STROBE (or appropriate STOBE extension) guideline (available from: https://www.equator-network.org/reporting-guidelines/strobe) and include the completed STROBE (or STROBE extension) checklist as Supporting Information. Please add the following statement, or similar, to the Methods: "This study is reported as per the Strengthening the Reporting of Observational Studies in Epidemiology (STROBE) guideline (S1 Checklist)." When completing the checklist, please use section and paragraph numbers, rather than page numbers.

* For all observational studies, in the manuscript text, please indicate: (1) the specific hypotheses you intended to test, (2) the analytical methods by which you planned to test them, (3) the analyses you actually performed, and (4) when reported analyses differ from those that were planned, transparent explanations for differences that affect the reliability of the study's results. If a reported analysis was performed based on an interesting but unanticipated pattern in the data, please be clear that the analysis was data driven.

* Please state in the Methods section whether the study had a prospective protocol or analysis plan. If a prospective analysis plan (from your funding proposal, IRB or other ethics committee submission, study protocol, or other planning document written before analyzing the data) was used in designing the study, please include the relevant document(s) with your revised manuscript as a Supporting Information file to be published alongside your study and cite it in the Methods section. A legend for this file should be included at the end of your manuscript. If no such document exists, please make sure that the Methods section transparently describes when analyses were planned, and when/why any data-driven changes to analyses took place. Changes in the analysis, including those made in response to peer review comments, should be identified as such in the Methods section of the paper, with rationale.

---

## [Decision Letter · Decision Letter 2]

26 Jun 2025

Dear Dr Marcuzzi,

Many thanks for submitting your manuscript "Parental opioid prescriptions and the risk of opioid use in adolescents and young adults: The HUNT Study linked with prescription registry data" (PMEDICINE-D-25-00353R2) to PLOS Medicine. The paper has been re-reviewed by one subject expert and a statistician; their comments are included below and can also be accessed here: [LINK]

As you will see, the reviewers think the manuscript is improved, but still have several remaining concerns. After discussing the paper with the editorial team and an academic editor with relevant expertise, we are inviting you to revise the paper to address these remaining comments. Please ensure that you address all the remaining concerns from the reviewers.

Specifically, we would like you to:

1) Address the comment from Reviewer #1, and ensure that the DAG is complete. You may also want to consider removing offspring BMI from the DAG.

2) Remove the analysis using BMI as a negative control, as BMI is a confounder.

3) Consider whether the conclusions can be generalizable. If not, please state this as a limitation in the discussion.

4) We are ok with parental BMI as a confounder (comment R2), but please ensure the explanation for this is clearly described in the manuscript.

We may send the revised paper to some or all of the original reviewers, and we cannot provide any guarantees at this stage regarding publication.

We ask that you submit your revision by Jul 17 2025 11:59PM. However, if this deadline is not feasible, please contact me by email, and we can discuss a suitable alternative.

Don't hesitate to contact me directly with any questions (sbruijn@plos.org).

Best regards,

Suzanne

Suzanne De Bruijn, PhD

Associate Editor

PLOS Medicine

sbruijn@plos.org

Comments from the reviewers:

Reviewer #1: I thank the authors for their considered replies. I have some remaining queries:

1. Further detail of follow-up is needed to ensure immortal time bias is not present and reverse causation. Parental opioid use after offspring participation (up to 3 years post - as per page 7 of methods) has a high-level potential for immortal time bias and reverse causation (exposure measured after the outcome). Please clarify.

2. Recognition the reimbursement codes were not available for all prescriptions as a limitation.

3. Imputation of BMI - was the adequacy of the imputation checked and assumptions of mode of missing checked prior? (ie missing at random).

4. Missig chronic pain - I agree that imputation would not be appropriate as it does not appear missing at random. Although the authors state there are no large differences this is not true: missing vs non missing for maternal opioid prescription; 1,559 (8.5%) vs 124 (6.5%) - Z test p-value; 0.018 and paternal 1,283 (8.2%) vs 125 (5.0%); p-value: <0.00001 and are likely not only statistically different but clinically meaningful. This should be noted in the results and discussion of this amended given the difference is not negligible.

5. The DAG is incomplete and does not consider the associations between the covariates. The inclusion of eg and multiple covariates is confusing. Chronic pain should be its own variable as an ancestor of exposure. Noting adjustment of the available covariates; Parental age, Parental education, parental BMI does seem sufficient to close biasing paths. However residual confounding is plausible, especially given other substance use data unavailable and should be noted as a limitation.

6. I do not agree with the use of BMI as a negative control; it is a confounder and clearly so in the DAG. A negative control for exposure must be a variable that does not affect the outcome.

Reviewer #2: Thank you for addressing my first set of comments. I appreciate the authors conducting a sensitivity analysis to address potential bias due to prevalent opioid use. However, some of the responses are inadequate or inaccurate.

For example, the idea that sample representativeness is not important in explanatory research is outdated. There is now robust literature that representativeness is important when there is treatment effect heterogeneity across sample characteristics, and those characteristics influence selection into the study sample. When both are true, estimates obtained in the selected sample will not generalize (i.e., transport) to the target population where clinical or policy decisions will be made (e.g., Hayes-Larson et al. 2024 Epidemiology). Additionally, the idea that results can be generalized to a wider population under the assumption of high internal validity is incorrect. A double-blind RCT with perfect adherence and no attrition would have high internal validity, but this does not guarantee its external validity (e.g., if trial participants are healthier, more motivated etc. than the general population.)

I'm still unconvinced that parental BMI is an appropriate confounder to adjust for for this research question. While the new DAG is helpful, it would suggest from the authors' explanation that offspring BMI is a descendent of parental BMI. If the authors truly believe that offspring BMI is a strong risk factor or cause of opioid prescriptions in offspring, then offspring BMI seems like the appropriate measure to adjust for, since it is unlikely to be an instrument whereas parental BMI could be.

Finally, the sentence in question in the Discussion is still poorly written. "…when prescribing opioids as part of managing pain conditions in adolescents and young adults" suggests to the reader that clinicians may need to be aware when "managing pain conditions [using opioids] in adolescents and young adults." Because the exposure was parental opioids, clinicians may need to be aware when using opioids to manage pain in parents.

---

*Non-technical Author summary:

-Please consider adding a sentence about the analysis performed to the "What did the researchers do and find?" section.

FIGURES AND TABLES

OBSERVATIONAL STUDIES

* For all observational studies, in the manuscript text, please indicate: (1) the specific hypotheses you intended to test, (2) the analytical methods by which you planned to test them, (3) the analyses you actually performed, and (4) when reported analyses differ from those that were planned, transparent explanations for differences that affect the reliability of the study's results. If a reported analysis was performed based on an interesting but unanticipated pattern in the data, please be clear that the analysis was data driven.

* Please state in the Methods section whether the study had a prospective protocol or analysis plan. If a prospective analysis plan (from your funding proposal, IRB or other ethics committee submission, study protocol, or other planning document written before analyzing the data) was used in designing the study, please include the relevant document(s) with your revised manuscript as a Supporting Information file to be published alongside your study and cite it in the Methods section. A legend for this file should be included at the end of your manuscript. If no such document exists, please make sure that the Methods section transparently describes when analyses were planned, and when/why any data-driven changes to analyses took place. Changes in the analysis, including those made in response to peer review comments, should be identified as such in the Methods section of the paper, with rationale.

---

## [Decision Letter · Decision Letter 3]

14 Aug 2025

Dear Dr. Marcuzzi,

Thank you very much for re-submitting your manuscript "Parental opioid prescriptions and the risk of opioid use in adolescents and young adults: The HUNT Study linked with prescription registry data" (PMEDICINE-D-25-00353R3) for review by PLOS Medicine.

I have discussed the paper with my colleagues and the academic editor and it was also seen again by 2 reviewers. I am pleased to say that provided the remaining editorial and production issues are dealt with we are planning to accept the paper for publication in the journal.

However, to address the remaining concern from one of the reviewers, as well as from the editors, we would like you to:

*We appreciate that you added the sensitivity analysis in Table S8. However, we would like to ask you to include this table in the main text. Furthermore, we would like you to provide a discussion of this table and the associated study limitation.

*Please add the numbers of the supplemental tables in the methods, where you describe these sensitivity analyses.

*if this is available, please provide more participant characteristics, for both parents and offspring. Including but not limited to income, education (more detailed than just % to 12 years), and ethnicity.

*Please include, either in the text or as a supplemental table, information about the timing of parental vs. offspring prescriptions.

*Clarify if there is any data on offspring opioid prescriptions at baseline.

********

We look forward to receiving the revised manuscript by Aug 21 2025 11:59PM.   

Sincerely,

Suzanne De Bruijn, PhD

Associate Editor 

PLOS Medicine

plosmedicine.org

Requests from Editors:

GENERAL EDITORIAL REQUESTS

* Please confirm that your title complies with to PLOS Medicine's style. Your title must be nondeclarative and not a question. It should begin with main concept if possible. "Effect of" should be used only if causality can be inferred, i.e., for an RCT. Please place the study design ("A randomized controlled trial," "A retrospective study," "A modelling study," etc.) in the subtitle (ie, after a colon).

* Please confirm that your abstract complies with our requirements, including format (three sections: Background, Methods and Findings, and Conclusions) and providing all the information relevant to this study type https://journals.plos.org/plosmedicine/s/submission-guidelines#loc-abstract

* Please ensure that the Introduction ends with a clear description of the study question or hypothesis.

* Please ensure that all abbreviations are defined at first use throughout the text.

* Please confirm that all numbers presented in the abstract are present and identical to numbers presented in the main manuscript text.

GENERAL

* Please remove the 'conclusions' subheading from the discussion. Please also remove any other subheadings from the discussion.

* Statistical reporting: Please revise throughout the manuscript, including tables and figures.

- Please report statistical information as follows to improve clarity for the reader ""22% (95% CI [13,28]; p</=)"".

- Please separate upper and lower bounds with commas instead of hyphens as the latter can be confused with reporting of negative values.

- Please repeat statistical definitions (HR, CI etc.) for each set of parentheses.

* In the abstract, please include the important dependent variables that are adjusted for in the analyses.

FUNDING STATEMENT

* The funding statement should include: specific grant numbers, initials of authors who received each award, URLs to sponsors’ websites. Also, please state whether any sponsors or funders (other than the named authors) played any role in study design, data collection and analysis, the decision to publish, or preparation of the manuscript. If they had no role in the research, include this sentence: “The funders had no role in study design, data collection and analysis, decision to publish, or preparation of the manuscript.

COMPETING INTERESTS STATEMENT

* All authors must declare their relevant competing interests per the PLOS policy, which can be seen here: https://journals.plos.org/plosmedicine/s/competing-interests For authors with ties to industry, please indicate whether any of the interests has a financial stake in the results of the current study.

FIGURES

* Please show graph axes beginning at zero. If this is not possible, please show a break in the axis.

OBSERVATIONAL, COHORT, CROSS-SECTIONAL, AND CASE CONTROL STUDIES

* Did your study have a prospective protocol or analysis plan? Please state this (either way) early in the Methods section.

* For all observational studies, in the manuscript text, please indicate: (1) the specific hypotheses you intended to test, (2) the analytical methods by which you planned to test them, (3) the analyses you actually performed, and (4) when reported analyses differ from those that were planned, transparent explanations for differences that affect the reliability of the study's results. If a reported analysis was performed based on an interesting but unanticipated pattern in the data, please be clear that the analysis was data-driven.

Comments from Reviewers:

Reviewer #1: I thank the authors for their considered replies and believe they have addressed most issues.

However, the concerns around reverse causation remain. If the editors are happy to accept parental opioid use without consideration to the temporal relationship with offspring use then I will not belabor the point.

Otherwise, I suggest removing those exposures that occurred after outcome. The aim of the study is to 'examine the association between parental opioid prescriptions and risk of opioid use in young people'. This cannot be assessed using an exposure (parental use) that captured after the outcome (child use). Supplemental Table 8 not address the concerns of reverse causation.

Additionally, the qualifier 'somewhat' on pages 10 and 18.

Reviewer #2: The authors have addressed my comments.

********

---

## [Editor Report · Decision Letter 4]

15 Sep 2025

Dear Dr Marcuzzi, 

On behalf of my colleagues and the Academic Editor, Alexander Tsai, I am pleased to inform you that we have agreed to publish your manuscript "Parental opioid prescriptions and the risk of opioid use in adolescents and young adults: The HUNT Study linked with prescription registry data" (PMEDICINE-D-25-00353R4) in PLOS Medicine.

As a last remaining editorial request, we like you to add the URL for the funder in your financial disclosure. Furthermore, before your manuscript can be formally accepted you will need to complete some formatting changes, which you will receive in a follow up email. Please be aware that it may take several days for you to receive this email; during this time no action is required by you. Once you have received these formatting requests, please note that your manuscript will not be scheduled for publication until you have made the required changes.

PRESS

Sincerely, 

Suzanne De Bruijn, PhD 

Associate Editor 

PLOS Medicine